

# Elevated serum FGF21 predicts the major adverse cardiovascular events in STEMI patients after emergency percutaneous coronary intervention

Lingyun Gu, Wenlong Jiang, Huidong Qian, Ruolong Zheng and Weizhang Li

Department of Cardiology, Jiangyin Hospital Affiliated to Southeast University, Jiangyin, Jiangsu, China

## ABSTRACT

**Background**. Although there have been several studies related to serum fibroblast growth factor 21 (FGF21) levels and acute myocardial infarction, the value of serum FGF21 levels in ST-segment elevation myocardial infarction (STEMI) patients after emergency percutaneous coronary intervention (PCI) has not been previously investigated.

**Methods**. A total of 348 STEMI patients who underwent emergency PCI were enrolled from January 2016 to December 2018. The primary endpoint was the occurrence of major adverse cardiovascular events (MACEs), with a median follow-up of 24 months. Eighty patients with stable angina (SA) who underwent selective PCI served as the control group. Serum FGF21 levels were measured by ELISA.

**Results**. Serum FGF21 levels were significantly higher in the STEMI group than in the SA group ($225.03 \pm 37.98$ vs. $135.51 \pm 34.48$, $P < 0.001$). Multiple linear regression analysis revealed that serum FGF21 levels were correlated with NT-proBNP ($P < 0.001$). According to receiver operating characteristic (ROC) analysis, the areas under the ROC curve (AUCs) of FGF21 and NT-proBNP were 0.812 and 0.865, respectively. The Kaplan-Meier curves showed that STEMI patients with lower FGF21 levels had an increased MACE-free survival rate. Cox analysis revealed that high FGF21 levels (HR: 2.011, 95% CI: [1.160–3.489]) proved to be a powerful tool in predicting the risk of MACEs among STEMI patients after emergency PCI.

**Conclusion**. Elevated FGF21 levels on admission have been shown to be a powerful predictor of MACEs for STEMI patients after emergency PCI.

## INTRODUCTION

Acute myocardial infarction (AMI) is one of the most common cardiovascular emergencies, with a high mortality and disability rate. An increasing number of studies have found that the inflammatory response following AMI plays a significant role in determining the infarct area and subsequent adverse left ventricular (LV) remodeling (*Ong et al., 2018*). As the understanding of inflammatory mechanisms following AMI has increased, many

Corresponding author
Weizhang Li, liwz@jyrmyy.com

liver-derived biomarkers with excellent prognostic value, such as C-reactive protein and interleukin-6, have been discovered (*Groot et al., 2015*; *Ziakas et al., 2006*).

Another liver-secreted cytokine with anti-inflammatory effects, fibroblast growth factor 21 (FGF21), has gained increasing interest in recent cardiovascular research. Previous studies have found that FGF21, in addition to its anti-inflammatory effects, has an antioxidative stress effect and regulates glycolipid metabolism (*Fisher & Maratos-Flier, 2016*; *Luo et al., 2017*). Previous studies have found that inflammation, oxidative stress, and disturbances in glycolipid metabolism play an important role in the pathogenesis of AMI (*Neri et al., 2017*; *Tao et al., 2015*; *Zhang et al., 2018*). Therefore, we speculate that FGF21 is correlated with the prognosis of AMI.

Previous studies have found that serum FGF21 levels were higher in patients with AMI than in patients with stable angina (SA), and correlated with the occurrence of major adverse cardiovascular events (MACEs) (*Chen, Lu & Zheng, 2018*; *Sunaga et al., 2019*; *Zhang et al., 2015*). However, these studies did not distinguish between STEMI patients and NSTEMI patients. Therefore, the relationship between serum FGF21 levels and the prognosis of STEMI patients, who account for 45.8% of AMI patients (*Kuch et al., 2007*), remains unclear. The goal of this study was to evaluate whether serum FGF21 levels on admission could predict MACEs in STEMI patients after emergency percutaneous coronary intervention (PCI).

## METHODS

### Study population

A total of 348 patients with STEMI admitted to Jiangyin People's Hospital were enrolled from January 2016 to December 2018. They all underwent emergency coronary angiography and PCI within 12 h after the onset of ischemic symptoms. The STEMI patients received medication according to the American College of Cardiology Foundation/American Heart Association (ACCF/AHA) guidelines (*O'Gara et al., 2013*). To exclude the influence of PCI on the experimental results, 80 patients with SA who underwent selective coronary angiography and PCI during the same period were selected as the control group.

The research was approved by the Ethics Committee of Jiangyin People's Hospital (approval number: 2015ER035), and written informed consent was obtained from all enrolled patients before participation. STEMI is a clinical syndrome defined by characteristic symptoms of myocardial ischemia in association with persistent electrocardiographic (ECG) ST elevation and the subsequent release of biomarkers of myocardial necrosis (*O'Gara et al., 2013*). Patients will be excluded if they present with the following diseases: cardiac shock, primary valvular heart disease, congenital heart disease, tachycardia-induced cardiomyopathy, pericardial disease, chronic liver insufficiency, acute renal failure, rheumatic disease, pulmonary embolism, neoplasm, inflammatory or infectious disorders, and excess alcohol consumption.

### Clinical and laboratory assessments

The baseline characteristics of the participants and history of hypertension, diabetes mellitus and atrial fibrillation were recorded by experienced physicians. All patients

underwent physical examination on admission, and elbow venous blood was drawn with a separation gel vacuum tube on an empty stomach early the next morning. A portion of the blood samples were sent to the central laboratory for troponin I testing by a Beckman DXI800 and for triglyceride, total cholesterol, low-density lipoprotein cholesterol (LDL-C), high-density lipoprotein cholesterol (HDL-C), uric acid, N-terminal proBNP (NT-proBNP) and creatinine assessment by Roche e602 and c701 modules, respectively. The other part of each blood sample was centrifuged to obtain serum and stored at −80 °C in a refrigerator, and serum FGF21 levels were measured every 3 months, as in our previous study (*Gu et al., 2021*).

## Echocardiography

All patients underwent echocardiography on 7-10 days after primary PCI (P-PCI). Echocardiogram was performed by experienced operators with an ultrasound machine (Philips iE 33 xMatrix).

The mitral regurgitation was examined and evaluated for severity using the flow convergence (PISA) method according to the guidelines of the American Society of echocardiography (*Zoghbi et al., 2003*). Pulmonary artery pressure was assessed by the tricuspid regurgitation method. The left atrial diameter (LAD), interventricular septum thickness (IVST), left ventricular posterior wall thickness (LVPWT), left ventricular end-diastolic diameter (LVEDD), left ventricular end-systolic diameter (LVESD) and left ventricular ejection fraction (LVEF) were measured in the parasternal long-axis view in the M-mode images. Left ventricular end-systolic volume (LVESV) and left ventricular end-diastolic volume (LVEDV) were measured by the Simpson method. Left ventricular mass (LVM) was calculated using the formula $0.8 \times 1.04 \times [(IVST + LVPWT + LVEDD)^3 - (LVEDD)^3] + 0.6$ (*Lang et al., 2015*).

## Follow-up

Follow-up, which was initiated at enrollment and ended in April 2020, was conducted for STEMI patients only. The median follow-up time was 24 months (interquartile range, 16-36.25 months). The primary endpoint of the study was the first MACE, defined as all-cause mortality or readmission for angina, heart failure, or recurrent AMI. The follow-up was conducted through periodic outpatient interviews and scripted telephone interviews with the patients or their families every month. Due to the high mobility of migrant workers and changes in phone numbers, 30 of the STEMI patients were lost to follow-up. Among all STEMI patients, 106 recorded MACEs, including 6 deaths and 100 readmissions (Fig. 1).

## Statistical analysis

Statistical analysis was conducted using the SPSS 22.0 statistical package. The quantitative variables are expressed as the means $\pm$ standard deviations and compared using Student's t test. Categorical variables were expressed as absolute numbers (percentages) and compared using the chi-square test and Mann–Whitney-Wilcoxon test. Multiple linear regression analysis was performed on the STEMI groups to evaluate the association between serum FGF21 concentration and other clinical covariates. Receiver operating characteristic (ROC)

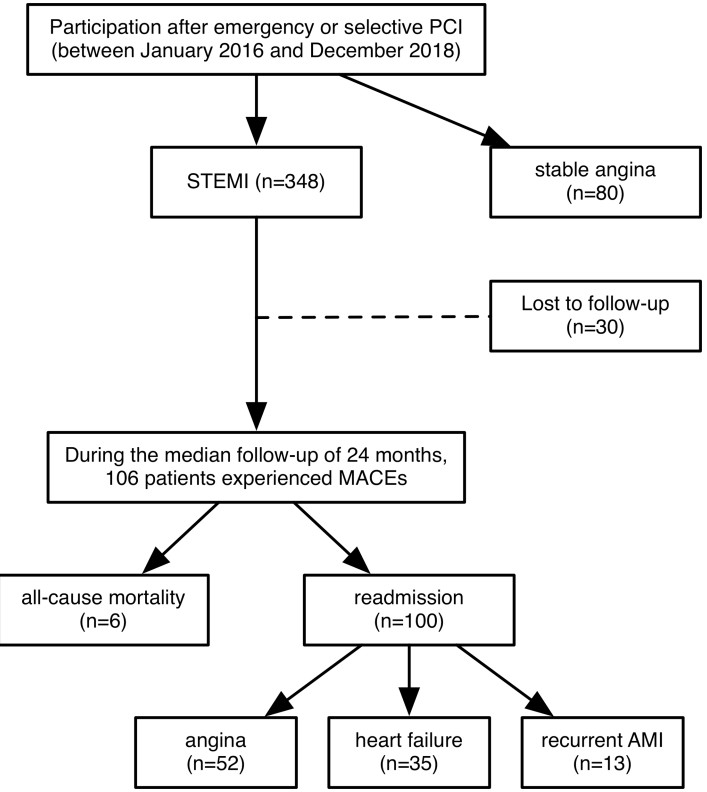

**Figure 1** **Flow chart of this study.** A total of 348 patients with STEMI were enrolled from January 2016 to December 2018. They all underwent emergency coronary angiography and PCI within 12 h after the onset of ischemic symptoms. 80 patients with stable angina who underwent selective coronary angiography and PCI were selected as the control group. Follow-up, which was initiated at enrollment and ended in April 2020, was conducted for STEMI patients only. The median follow-up time was 24 months (interquartile range, 16–36.25 months). The primary endpoint of the study was the first MACE, defined as all-cause mortality or readmission for angina, heart failure, or recurrent AMI. Eventually, 30 of the STEMI patients were lost to follow-up and 106 recorded MACEs, including 6 deaths and 100 readmissions. PCI: percutaneous coronary intervention; STEMI: st-segment elevation myocardial infarction; MACEs: major adverse cardiovascular events; AMI: acute myocardial infarction.

analysis was used to determine the accuracy of the best cutoff value of variables for predicting the occurrence of MACEs. MACE-free survival analysis was performed with Kaplan–Meier survival curves and compared using the log-rank test. Univariate and multivariate Cox proportional hazards models were used to analyze the relationships between FGF21 levels and MACEs by calculating hazard ratios (HRs) and 95% CIs. Multivariate Cox analysis was performed on all variables that reached statistical significance in univariate Cox analysis. A $p$ value<0.05 was considered statistically significant (two-tailed).

## RESULTS

The demographic and baseline clinical characteristics of the subjects are summarized in Table 1. No differences were observed in sex, age, incidence of atrial fibrillation, levels of triglycerides, total cholesterol, uric acid, creatinine, pulmonary pressure, mitral

regurgitation, IVST, LVPWT and LVEDV between the STEMI and SA groups. Compared with the SA group, the STEMI group had a higher incidence of hypertension and diabetes. LDL-C and NT-proBNP levels were markedly higher in the STEMI group than in the SA group, while the HDL-C levels were lower in the STEMI group than in the SA group (all $P < 0.05$). The LAD, LVEDD, LVESD, LVESV, LVM and LVEF were also markedly different between the groups. There were significant differences between the two groups in the culprit artery and lesion site ($P < 0.05$). The levels of serum FGF21 were significantly higher in the STEMI group than in the SA group (225.03 $\pm$ 37.98 *vs.* 135.51 $\pm$ 34.48, $P < 0.001$) (Table 1). Multiple linear regression analysis revealed that serum FGF21 levels were correlated with NT-proBNP levels ($P < 0.001$).

During the median follow-up of 24 months, 30 patients (8.62%) were lost to follow-up.106 patients (30.46%) experienced MACEs in the study: 6 patients died, and 100 patients were rehospitalized due to angina, heart failure or recurrent AMI. According to the ROC analysis, both FGF21 and NT-proBNP levels were significant indicators of MACEs in STEMI patients (Fig. 2). The area under the ROC curve (AUC) of NT-proBNP was 0.865 (95% CI [0.822–0.909]), which was greater than that of FGF21 (AUC, 0.812; 95% CI [0.765–0.860]). The difference between the two AUCs was statistically significant ($Z = 2.230$, $P = 0.026$).

According to the optimal cutoff value determined by ROC analysis and Youden index, FGF21 was divided into high FGF21 level group and low FGF21 level group for Kaplan–Meier analysis. The optimal cutoff value of FGF21 was 229.77 pg/ml, with a corresponding sensitivity of 84.00% and a specificity of 65.10%.

In the Kaplan–Meier analysis, a significant difference was observed between the high serum FGF21 level group and the low serum FGF21 level group (Fig. 3). The Mace-free probability among STEMI patients with high serum FGF21 levels was significantly lower than that of STEMI patients with low serum FGF21 levels.

The results of univariate and multivariate Cox proportional hazards models are shown in Table 2. Univariate Cox analysis showed that a high FGF21 level was a predictor for MACEs in STEMI patients, in addition to age, creatinine, troponin I, NT-proBNP, pulmonary pressure, mitral regurgitation, LAD, LVESD, LVESV and LVEF. In the multivariate Cox analysis, high FGF21 levels (HR: 2.224, 95% CI [1.122–4.407]) also proved to be a powerful tool in predicting the risk of MACEs. We also found that troponin I (HR: 6.842, 95% CI [3.870–12.094]), NT-proBNP levels (HR:3.452, 95% CI [1.765–6.750]), and mitral regurgitation (HR: 2.011, 95% CI [1.160–3.489]) increased the risk of MACEs.

## DISCUSSION

In this study, we demonstrated that serum FGF21 levels were significantly elevated in STEMI patients and correlated with NT-proBNP. At a median follow-up of 24 months, MACE-free survival was reduced in STEMI patients with elevated FGF21 levels, and elevated FGF21 levels were shown to be a strong predictor of MACEs in STEMI patients.

Recent studies have found that the heart is not only a target organ for FGF21 but also one of its sources (*Planavila et al., 2015*). The expression of FGF21 increases due to stimulation

**Table 1 The demographic and baseline clinical characteristics of the patients.**

| Variables | STEMI group (*n* = 348) | SA group (*n* = 80) | *P* value |
|---|---|---|---|
| Demographic data | | | |
|     Male, n (%) | 280 (80.46%) | 57 (71.25%) | 0.069 |
|     Age (years) | 62.05 ± 13.05 | 59.70 ± 12.56 | 0.144 |
|     Hypertension, n (%) | 185(53.16%) | 17 (21.25%) | 0.020 |
|     Diabetes mellitus, n (%) | 64 (18.39%) | 7 (8.75%) | 0.037 |
|     Atrial fibrillation, n (%) | 27 (7.76%) | 4 (5.00%) | 0.391 |
| Laboratory data | | | |
|     Total cholesterol (mmol/L) | 4.33 ± 1.11 | 4.50 ± 1.29 | 0.271 |
|     Triglyceride (mmol/L) | 1.83 ± 1.44 | 2.04 ± 1.57 | 0.245 |
|     LDL-C (mmol/L) | 3.26 ± 1.01 | 2.49 ± 0.72 | 0.000 |
|     HDL-C (mmol/L) | 1.14 ± 0.44 | 1.29 ± 0.42 | 0.008 |
|     Uric acid (μmol/L) | 340.16 ± 98.12 | 332.52 ± 59.90 | 0.370 |
|     Creatinin (μmol/L) | 80.88 ± 59.96 | 72.18 ± 20.05 | 0.219 |
|     Troponin I (ng/ml) | 44.07 ± 20.76 | – | – |
|     NT-proBNP (pg/ml) | 593.17 ± 584.25 | 164.03 ± 100.25 | 0.000 |
|     FGF21 (pg/ml) | 225.03 ± 37.98 | 135.51 ± 34.48 | 0.000 |
| Echocardiographic data | | | |
|     Pulmonary pressure (mmHg) | 32.30 ± 8.34 | 32.00 ± 6.16 | 0.762 |
|     Mitral regurgitation | 49 (14.08%) | 11 (13.75%) | 0.818 |
|     LAD (mm) | 41.26 ± 4.74 | 39.68 ± 4.05 | 0.006 |
|     IVST (mm) | 9.75 ± 1.69 | 9.63 ± 1.33 | 0.548 |
|     LVPWT (mm) | 9.86 ± 1.35 | 9.65 ± 1.25 | 0.203 |
|     LVEDD (mm) | 52.42 ± 4.45 | 50.33 ± 3.96 | 0.000 |
|     LVESD (mm) | 37.75 ± 4.83 | 33.18 ± 4.07 | 0.000 |
|     LVEDV (ml) | 127.18 ± 38.03 | 121.08 ± 21.89 | 0.057 |
|     LVESV (ml) | 59.63 ± 23.52 | 45.76 ± 14.58 | 0.000 |
|     LVM (g) | 193.21 ± 42.47 | 177.35 ± 40.14 | 0.003 |
|     LVEF (%) | 53.66 ± 7.99 | 62.36 ± 6.32 | 0.000 |
| Angiographic data | | | |
|     Culprit artery | | | 0.003 |
|         LAD, n (%) | 207 (59.48%) | 61 (76.25%) | |
|         LCX, n (%) | 32 (9.20%) | 7 (8.75) | |
|         RCA, n (%) | 109 (31.32%) | 12 (15.00%) | |
|     Culprit lesion site | | | 0.038 |
|         Proximal segment, n (%) | 195 (56.03%) | 32 (40.00%) | |
|         Middle segment, n (%) | 121 (34.77%) | 43 (53.75%) | |
|         Distal segment, n (%) | 32 (9.20%) | 5 (6.25) | |

**Table 1** (*continued*)

| Variables | STEMI group (n = 348) | SA group (n = 80) | P value |
|---|---|---|---|
| Number of diseased vessels | | | 0.194 |
| One vessels, n (%) | 248 (71.26%) | 62 (77.50%) | |
| Two vessels, n (%) | 67 (19.25%) | 15 (18.75%) | |
| Three vessels, n (%) | 33 (9.48%) | 3 (3.75%) | |

**Notes.**

LDL-C, low density lipoprotein cholesterol; HDL-C, high density lipoprotein cholesterol; NT-proBNP, N-terminal proB-type natriuretic peptide; FGF21, fibroblast growth factor 21; IVST, interventricular septal wall thickness; LVPWT, left ventricular posterior wall thickness; LVEDD, left ventricular end-diastolic diameter; LVESD, left ventricular systolic diameter; LVEDV, left ventricular end-diastolic volume; LVESV, left ventricular end-systolic volume; LVM, left ventricular mass; LVEF, left ventricular ejection fraction; LAD, left atrial dimension; LCX, left circumflex artery; RCA, right coronary artery.

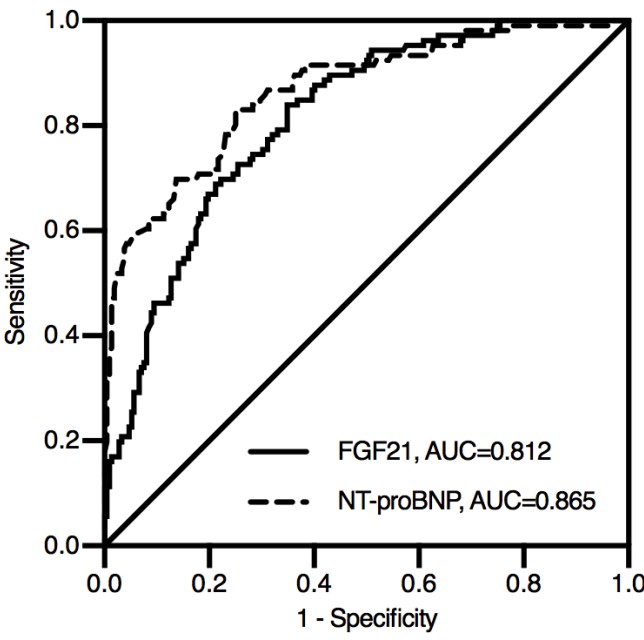

**Figure 2** **Receiver operating characteristic curve for predicting prognosis in STEMI patients after emergency PCI.** During the median follow-up of 24 months, 106 patients experienced MACEs in the study: 6 patients died, and 100 patients were rehospitalized due to angina, heart failure or recurrent AMI. According to the ROC analysis, both FGF21 and NT-proBNP levels were significant indicators of MACEs in STEMI patients. The area under the ROC curve (AUC) of NT-proBNP was 0.865 (95% CI [0.822–0.909]), which was greater than that of FGF21 (AUC, 0.812; 95% CI [0.765–0.860]). The difference between the two AUCs was statistically significant ($Z = 2.230$, $P = 0.026$). NT-proBNP, N-terminal proB-type natriuretic peptide; FGF21, fibroblast growth factor 21.

by inflammation, oxidative stress, lipid toxicity and endoplasmic reticulum stress, which plays an important role in cardiovascular disease (*Brahma et al., 2014*; *Planavila et al., 2015*; *Steven et al., 2019*). Then, FGF21 is synthesized by cardiomyocytes and secreted in an autocrine manner (*Planavila et al., 2013*; *Planavila, Redondo-Angulo & Villarroya, 2015*). In target cells, FGF21 binds to FGF receptors and $\beta$ Klotho, which are widely expressed in cardiomyocytes (*Domouzoglou et al., 2015*). Increased cardiac FGF21 expression protects

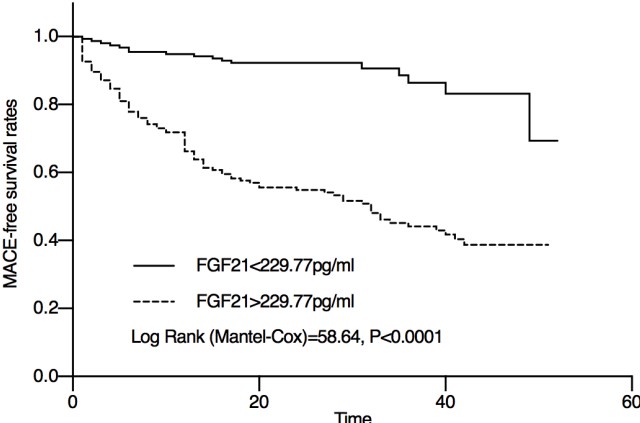

**Figure 3** **Kaplan–Meier curve showing the Mace-free probability of STEMI patients after emergency PCI with different FGF21 levels.** According to the optimal cutoff value determined by ROC analysis and Youden index, FGF21 was divided into high FGF21 level group and low FGF21 level group. The optimal cutoff value of FGF21 was 229.77 pg/ml. The Mace-free probability among STEMI patients with high serum FGF21 levels was significantly lower than that of STEMI patients with low serum FGF21 levels.

cardiomyocytes from inflammation, oxidative stress and dysfunction after myocardial I/R injury (*Hu et al., 2018*; *Zhang et al., 2016*).

Accumulating evidence indicates that elevated FGF21 concentrations are associated with a higher incidence of diabetes, chronic kidney disease, hypertension, and cardiovascular diseases, including coronary artery disease, heart failure, atrial fibrillation and AMI (*Gu et al., 2021*; *Shen et al., 2017*; *Shen et al., 2018*; *Wang et al., 2015*; *Wu et al., 2018*; *Zhang et al., 2015*). Moreover, FGF21 significantly predicts the incidence of coronary artery disease and all-cause and cardiovascular mortality (*Lakhani et al., 2018*). This is consistent with the results of our study.

However, the increase in FGF21 levels, which has a cardioprotective effect, is negatively correlated with the prognosis of cardiovascular disease. Previous research reveals that the elevation of FGF21 levels in pathological conditions may be part of a compensatory response to the underlying metabolic stress or may be caused by FGF21 resistance as a result of impaired FGF21 signaling (*Lewis et al., 2019*; *Woo et al., 2013*). Therefore, more FGF21 is secreted to compensate for its reduced metabolic regulation caused by FGF21 resistance. Moreover, additional supplementation with FGF21 has been shown to have a therapeutic effect in *in vitro* trials (*Zhu et al., 2014*). Furthermore, some companies, such as Eli Lilly and Pfizer, have now developed FGF21 analogs and demonstrated their therapeutic potential in clinical trials for the treatment of metabolic diseases (*BonDurant & Potthoff, 2018*).

Inconsistent with previous studies, the Multi-Ethnic Study of Atherosclerosis (MESA) recently found that FGF21 levels were not associated with incident cardiovascular disease after multiple adjustments (*Ong et al., 2019*); however, this study enrolled only patients without cardiovascular disease. In contrast to MESA, the patients enrolled in our study were STEMI patients, and we found that FGF21 was associated with the occurrence of

**Table 2  Univariate and multivariate COX analysis for prognosis in STEMI patients.**

| Variables | Univariate analysis | | | Multivariate analysis | | |
|---|---|---|---|---|---|---|
| | HR | 95%CI | *P* value | HR | 95%CI | *P* value |
| Sex (male) | 0.744 | 0.474-1.166 | 0.197 | – | – | – |
| Age (years) | 1.766 | 1.188-2.626 | 0.005 | – | – | – |
| Hypertension | 1.102 | 0.752-1.616 | 0.619 | – | – | – |
| Diabetes mellitus | 1.398 | 0.874-2.236 | 0.162 | – | – | – |
| Atrial fibrillation | 1.317 | 0.705-2.461 | 0.388 | – | – | – |
| Total cholesterol | 0.872 | 0.728-1.045 | 0.139 | – | – | – |
| Triglyceride | 0.850 | 0.718-1.006 | 0.059 | – | – | – |
| LDL-C | 1.168 | 0.965-1.414 | 0.111 | – | – | – |
| HDL-C | 0.773 | 0.479-1.245 | 0.289 | – | – | – |
| Uric acid | 1.000 | 0.998-1.001 | 0.676 | – | – | – |
| Creatinine | 1.002 | 1.000-1.004 | 0.016 | – | – | – |
| Troponin I | 8.143 | 4.834-14.716 | 0.000 | 6.842 | 3.870-12.094 | 0.000 |
| NT-proBNP | 7.496 | 4.508-12.462 | 0.000 | 3.452 | 1.765-6.750 | 0.000 |
| FGF21 | 5.901 | 3.509-9.922 | 0.000 | 2.224 | 1.122-4.407 | 0.022 |
| Pulmonary pressure | 1.510 | 1.015-2.247 | 0.042 | – | – | – |
| Mitral regurgitation | 1.745 | 1.071-2.844 | 0.025 | 2.011 | 1.160-3.489 | 0.013 |
| LAD | 1.732 | 1.178-2.547 | 0.005 | – | – | – |
| IVST | 1.381 | 0.671-2.842 | 0.381 | – | – | – |
| LVPWT | 0.987 | 0.662-1.471 | 0.948 | – | – | – |
| LVEDD | 1.359 | 0.924-1.998 | 0.119 | – | – | – |
| LVESD | 1.749 | 1.170-2.614 | 0.006 | – | – | – |
| LVEDV | 1.359 | 0.924-1.998 | 0.119 | – | – | – |
| LVESV | 1.749 | 1.170-2.614 | 0.006 | – | – | – |
| LVM | 1.334 | 0.882-2.016 | 0.172 | – | – | – |
| LVEF | 1.999 | 1.339-2.985 | 0.001 | – | – | – |
| Culprit artery | | | | | | |
| LAD | | | 0.077 | – | – | – |
| LCX | 0.487 | 0.212-1.120 | 0.091 | – | – | – |
| RCA | 0.671 | 0.426-1.0574 | 0.085 | – | – | – |
| Culprit lesion site | | | | | | |
| Proximal segment | 4.095 | 1.287-13.028 | 0.017 | – | – | – |
| Middle segment | 3.660 | 1.127-11.892 | 0.031 | – | – | – |
| Distal segment | | | 0.056 | – | – | – |
| Number of diseased vessels | | | | | | |
| One vessels | | | 0.186 | – | – | – |
| Two vessels | 1.144 | 0.921-2.262 | 0.109 | – | – | – |
| Three vessels | 0.794 | 0.383-1.649 | 0.794 | – | – | – |

**Notes.**

LDL-C, low density lipoprotein cholesterol; HDL-C, high density lipoprotein cholesterol; NT-proBNP, N-terminal proB-type natriuretic peptide; FGF21, fibroblast growth factor 21; IVST, interventricular septal wall thickness; LVPWT, left ventricular posterior wall thickness; LVEDD, left ventricular end-diastolic diameter; LVESD, left ventricular systolic diameter; LVEDV, left ventricular end-diastolic volume; LVESV, left ventricular end-systolic volume; LVM, left ventricular mass; LVEF, left ventricular ejection fraction; LAD, left atrial dimension; LCX, left circumflex artery; RCA, right coronary artery.

MACEs. In MESA, 481 patients underwent PCI because of cardiac ischemia, mainly acute coronary syndrome (ACS). Subgroup analysis showed that the higher the FGF21 level was, the sooner the ACS occurred and the higher the incidence of PCI. These studies suggested that for patients with cardiovascular disease, elevated FGF21 levels indicated a poor prognosis. Similarly, serum FGF21 levels were an independent predictor of incident coronary heart disease (CHD) in diabetic patients (*Lee et al., 2017*). In general, for patients with cardiovascular disease or diabetes, FGF21 is likely to be a powerful biomarker for predicting MACEs. However, for healthy people, a large number of studies are still needed to validate this conclusion.

Although several studies have focused on the association of FGF21 with AMI, we found for the first time that FGF21 was related to the prognosis of STEMI patients after treatment with emergency PCI. Sunaga et al. found that serum FGF21 levels were significantly higher in patients with AMI after PCI than in patients with SA, but this study was not followed up (*Sunaga et al., 2019*). Another study that included a sample of 55 STEMI and NSTEMI patients found that FGF21 was associated with AMI and with the incidence of reinfarction within 30 days after onset (*Zhang et al., 2015*). Similar to our study, Chen et al. found that FGF21 was associated with MACEs in patients with AMI (*Chen, Lu & Zheng, 2018*). However, unlike our study, the study did not distinguish between STEMI patients and NSTEMI patients, although STEMI patients were enrolled in the study. Overall, FGF21 was first identified as a MACE-related biomarker in our follow-up study of STEMI patients treated with emergency PCI.

The study had several limitations. Our study was conducted at a single hospital and included only Chinese people. Although our sample size meets the needs of the study, our follow-up time was too short, and the number of endpoints was not large enough. The absence of height, weight, ECG and peak troponin I data made it impossible to assess other factors associated with MACEs. Unfortunately, we did not follow up the SA group and therefore could not compare it to the STEMI group. In addition, serum FGF21 levels were measured only at admission instead of including a repeated measurement at the time of readmission. Therefore, large-scale and long-term clinical studies are needed to confirm the role of FGF21 and its relationship to STEMI.

Moreover, early echocardiography was used to assess the prognosis of STEMI patients who underwent emergency PCI. Although early echocardiography is not the best method to predict outcomes, some studies show that early echocardiography has a predictive value for the prognosis of AMI (*Awad et al., 2020*; *Cetin et al., 2021*; *Reindl et al., 2019*). However, a better approach would be to perform an echocardiography at the follow-up visit with a before-and-after comparison.

## CONCLUSIONS

In conclusion, we observed that serum FGF21 levels were significantly elevated in STEMI patients who had undergone emergency PCI. Moreover, elevated FGF21 levels on admission have been shown to be a powerful predictor of MACEs. Therefore, FGF21 can be used as a novel biomarker for STEMI after emergency PCI.

## ACKNOWLEDGEMENTS

The author thanks Professor Yao Yuyu for his assistance in this manuscript.

### Funding

The authors received no funding for this work.

### Competing Interests

The authors declare there are no competing interests.

### Author Contributions

- Lingyun Gu conceived and designed the experiments, performed the experiments, analyzed the data, prepared figures and/or tables, and approved the final draft.
- Wenlong Jiang conceived and designed the experiments, prepared figures and/or tables, authored or reviewed drafts of the paper, and approved the final draft.
- Huidong Qian performed the experiments, prepared figures and/or tables, and approved the final draft.
- Ruolong Zheng conceived and designed the experiments, authored or reviewed drafts of the paper, and approved the final draft.
- Weizhang Li analyzed the data, authored or reviewed drafts of the paper, and approved the final draft.

### Human Ethics

The following information was supplied relating to ethical approvals (i.e., approving body and any reference numbers):

The Ethics Committee of Jiangyin People's Hospital approved this research (2015ER035).

### Data Availability

The raw measurements are available in the Supplementary File.

### Supplemental Information

Supplemental information for this article can be found online at http://dx.doi.org/10.7717/peerj.12235#supplemental-information.

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
