# Peer review of "Elevated serum FGF21 predicts the major adverse cardiovascular events in STEMI patients after emergency percutaneous coronary intervention"

_PeerJ, doi:10.7717/peerj.12235_

## Round 0.1 · original submission · Major Revisions

New parameters should be taken into account, as suggested by reviewer 1, and probably the statistical analysis should be extended accordingly.
In addition, the certainty tone of discussion and conclusions regarding the role of FGF21 as biomarker of future MACE needs to be lessened.

·

Basic reporting

Elevated serum FGF21 predicts the major adverse cardiovascular events in STEMI patients after emergency percutaneous coronary intervention

Review
The authors try to verify if FGF21 may be a marker for severity in AMI patients.
And they concluded that over 240 pg/ml FGF21 identifies some higher risk AMI patients.

I have some QUESTIONS regarding the study:

1) Why they didn’t perform a FUP in patients with stable angina, which was considered by the authors as “control group”? Being a “control group”, they would have the possibility to assess the ROC also in the stable angina group, because the level of FGF21 was not zero in this group. Is it possible to know how many patients have had multivessel coronary disease in this stable angina patients? In this manner will be possible to compare better the two groups of patients.
2) It is not clear for me way the authors choose to confront only FGF21 level with MACE, neglecting many others important factor with negative prognosis after AMI…
3) Why the authors don’t try to quantify the dimension of AMI using ECG criteria, as QS aspect by the way. In this manner will be possible to rule out one of important element of MACE, which is the big AMI. Or to assess the correlation of this one with MACE and FGF21.
4) We do not have also the maximum level of troponin at the moment of acute MI… Why?
5) In a group of 348 pts with AMI, why we cannot have information about the localization of the AMI. Closure of left anterior descending artery and anterior AMI is quite a different disease comparing with an inferior AMI, in case of closure of right coronary artery… So, the MACE of these group of patients will be different… Does the localization of AMI correlate with FGF21 and MACE?
6) Why the authors don’t specify the position of culprit lesion on angiography? Proximal one? Distal one? Does the position of culprit lesion correlate with FGF21 and MACE?
7) What has been the burden of coronary disease at angiography study? How many patients have had one vessel, two or three vessel disease at coronary angiography? Does the burden of coronary artery disease in AMI group correlates with FGF21 and MACE?
8) The authors choose to assess the consequences of AMI performing an echocardiographic exam in the first 7-10 days after the acute event. The phenomenon of post infarction remodeling is a long one and only a single Echo exam, performed after 7-10 days may not reflects it. So, use of Echo as method the evaluate the extension and the gravity of AMI at 7/10 days is not sufficient… Or is too early… Way we cannot see the echo exam on FUP?
9) In a group of 348 pts with AMI how many patients have had mitral regurgitation? What about the degree of MR? Does the MR correlates with MACE and FGF21?
10) What about the systolic pulmonary pressure in patients with MACE? Does the pulmonary pressure correlates with MACE and FGF21?
11) Why they used only left ventricular end diastolic diameter (LV EDD) and only the ejection fraction to assess the degree of LV remodeling and the level of cardiac reserve after AMI. The LV volumes express much better the phenomenon of post myocardial infarction than LV EDD. Does the LV indexed volumes correlate with MAXCE and FGF21?
12) The authors have to clarify how LV EDD diameter was measured: in M mode or in 2DE? And in which view: PS Long Axis? PS Short Axis?
13) Consideration: the cardiac reserve is better expressed by the end systolic diameter, and from the study does not appear that they use this parameter (which is by far more characteristic for reduced cardiac reserve). If the authors don’t assess it, they have to specify why. Does the end systolic diameter correlates with MACE and FGF21?
14) The same consideration for LVEF: end systolic volumes (indexed) correlates better with a poor cardiac reserve, that EF itself. The authors have to publish the indexed volumes of LV cavity and also the methods of calculation of these volumes: may be Simpson?
15) Did the authors assess the Left Atrial indexed volumes (in A4C and A2C view)? Does the indexed LA volume correlate with MACE and FGF21?
16) If they performed the M mode assessment, what about the calculation LV mass by this modality? There is any correlation with MACE? Usually it is considered that the increased LV mass predicts heart failure. Does, in this group of patients, LV mass correlate with MACE and FGF21?
17) How many patients in 348 AMI patients presented atrial fibrillation, or other arrhythmias at the onset of AMI? What has been the impact of arrhythmias on MACE and FGF21?

In conclusion

The future of AMI patient depends of the amount of necrosis and the residual ischemic burden. As known, when the amount of necrotic tissue is significant / important, the LV will suffer a phenomenon cavity of remodeling.
Consideration: only EF and LE EDD, assessed after 7-10 days post AMI may not express enough the degree of post AMI remodeling. Based on these criteria, the use of echocardiography exam at 7/10 days after AMI to identify the “bad patients”, is not sufficient and is not a valid solution.
In the group of study, we do not have important information to characterize the entity and the amount of necrotic tissue. Does not influence the amount of necrotic tissue the level of FGF21 and MACE?
The authors must specify the position of culprit lesion, the numbers of ECG derivation with QS aspect, the localization of AMI and to specify the maximum level of troponin at the onset of AMI.
The absence of data regarding LV and LA indexed volumes, mitral regurgitation and pulmonary pressure do not allow to rule out other very important factors in MACE, and confront them with FGF21.
The myocardial infarction is a very complex disease and phenomenon; use of FGF21 only to identify MACE appears as a limited and confounding method.
Presented in this isolated / detached manner, seems that the only element able to identify MACE may be FGF21…
If this is the message / aim, they have to exclude the echocardiography and many others presented data.

Yours
Gheorghe CERIN MD, PhD, FESC

Experimental design

See the BASIC REPORTING please!

Validity of the findings

See the BASIC REPORTING please!

Additional comments

In conclusion

The future of AMI patient depends of the amount of necrosis and the residual ischemic burden. As known, when the amount of necrotic tissue is significant / important, the LV will suffer a phenomenon cavity of remodeling.
Consideration: only EF and LE EDD, assessed after 7-10 days post AMI may not express enough the degree of post AMI remodeling. Based on these criteria, the use of echocardiography exam at 7/10 days after AMI to identify the “bad patients”, is not sufficient and is not a valid solution.
In the group of study, we do not have important information to characterize the entity and the amount of necrotic tissue. Does not influence the amount of necrotic tissue the level of FGF21 and MACE?
The authors must specify the position of culprit lesion, the numbers of ECG derivation with QS aspect, the localization of AMI and to specify the maximum level of troponin at the onset of AMI.
The absence of data regarding LV and LA indexed volumes, mitral regurgitation and pulmonary pressure do not allow to rule out other very important factors in MACE, and confront them with FGF21.
The myocardial infarction is a very complex disease and phenomenon; use of FGF21 only to identify MACE appears as a limited and confounding method.
Presented in this isolated / detached manner, seems that the only element able to identify MACE may be FGF21…
If this is the message / aim, they have to exclude the echocardiography and many others presented data.

Yours
Gheorghe CERIN MD, PhD, FESC

Reviewer 2 ·

Basic reporting

The article is written in a clear manner, conforming to professional standards.

Experimental design

The authors proposed to research the link between FGF21 levels and the risk of MACE in myocardial infarction. The theme is appealing and the authors state in the background section that this relationship has not been previously investigated. However, in the introduction section they cite several studies that have analyzed this subject, so rephrasing the background section would be beneficial. The investigation is rigorous, but there is a discrepancy in the number of patients included in the control group (80) and the STEMI group (348). Even more, the follow-up study was done only in the STEMI group. A patients’ consent translated in english, attached to the original one would be recommended.

Validity of the findings

The findings are well presented and the statistical analysis is solid, but I would encourage the authors to develop the section regarding the echo findings in relation to the analyzed marker, or to remove it.

---

## Round 0.2 · Major Revisions

Please carefully take into account the comments from reviewer 1 and add to the manuscript appropriate changes and/
or comments.
Some mandatory points to be addressed:

- Comments about the limited prognostic utility of early echocardiography (in the first ten days of myocardial infarction) to predict outcomes;

- Clarify the criteria for ischemic MR quantification used in the study;

- Clarify the NYHA class;

- Add to the study limitations the lack of information on weight, height, and extension of AMI based on ECG.

·

Basic reporting

.

Experimental design

.

Validity of the findings

POINT-TO-POINT REPLY TO THE REVIEWER’ COMMENTS

We would like to thank the reviewers for taking their time to review our manuscript. Their comments have greatly helped us strengthened this manuscript and revisions have been made to address the reviewers’ concerns as well as improve on the clarity and focus of the manuscript. Content highlighted in red are the revisions that have been made in the manuscript.

Reviewer 1 (Gheorghe Cerin)
1) Why they didn’t perform a FUP in patients with stable angina, which was considered by the authors as “control group”? Being a “control group”, they would have the possibility to assess the ROC also in the stable angina group, because the level of FGF21 was not zero in this group. Is it possible to know how many patients have had multivessel coronary disease in this stable angina patients? In this manner will be possible to compare better the two groups of patients.
Reply: Thank you very much for reviewer's valuable advice. Because we initially designed the trial to focus only on the prognostic value of FGF21 in STEMI, we did not follow up patients with stable angina. We are very sorry for the confusion caused by the negligence of the experimental design. The coronary angiography data of patients with stable angina have been added to the manuscript by reviewing the case data.

Comment: may be in this case the word “control” is not suitable and has to be changed…

2) It is not clear for me way the authors choose to confront only FGF21 level with MACE, neglecting many others important factor with negative prognosis after AMI…
Reply: We feel very sorry for the confusion caused by ignoring other prognostic factors. Because of the physiological role of FGF21 and our previous studies, we only focused on the relationship between FGF21 and MACE. Other clinical data related to MACE, especially echocardiography data and coronary angiography data have been supplemented in the article.

No comment.

3) Why the authors don’t try to quantify the dimension of AMI using ECG criteria, as QS aspect by the way. In this manner will be possible to rule out one of important element of MACE, which is the big AMI. Or to assess the correlation of this one with MACE and FGF21.
Reply: Thank you very much for the valuable suggestions of the reviewer, and we apologize for ignoring the ECG data. We will pay attention to ECG data in the subsequent study of AMI.

No comment.

4) We do not have also the maximum level of troponin at the moment of acute MI… Why?
Reply: All patients underwent physical examination on admission and elbow venous blood was drawn with a gel tube on an empty stomach early the next morning. Because the diagnosis of the patient is clear, we don't test troponin I level frequently. We actually test troponin I every 2-3 days. Based on the dynamic evolution of troponin, in fact, the troponin level we tested on the second day of admission was at the highest level during hospitalization. This was confirmed by consulting the patient’s hospitalization data.

Comment - The problem remains anyway… Bigger will the AMI, higher will be the probability to develop MACE… Doesn’t matter so much if the troponin was not measured in the first o in the second day after AMI…

5) In a group of 348 pts with AMI, why we cannot have information about the localization of the AMI. Closure of left anterior descending artery and anterior AMI is quite a different disease comparing with an inferior AMI, in case of closure of right coronary artery… So, the MACE of these group of patients will be different… Does the localization of AMI correlate with FGF21 and MACE?
Reply: Thank you very much for the valuable suggestions of the reviewer. By consulting the inpatient data, we have added the data of coronary angiography to this article. Through statistical analysis, we found that culprit artery was not associated with MACE.
Unusual on my opinion... Occlusion of the left anterior descending coronary artery is a very different clinical picture and disease, as we all know, compared with the occlusion of the right coronary artery… I consider that an explanation in the discussion chapter may be necessary. Probably in this case we have to consider the benefic effect of rescue PTCA probably.

No comment.

6) Why the authors don’t specify the position of culprit lesion on angiography? Proximal one? Distal one? Does the position of culprit lesion correlate with FGF21 and MACE?
Reply: Thank you very much for the valuable suggestions of the reviewer. By consulting the inpatient data, we have added the data of coronary angiography to this article. Through statistical analysis, we found that culprit lesion site was associated with MACE.

No comment.

7) What has been the burden of coronary disease at angiography study? How many. patients have had one vessel, two or three vessel disease at coronary angiography? Does the burden of coronary artery disease in AMI group correlates with FGF21 and MACE?
Reply: Thank you very much for the valuable suggestions of the reviewer. By consulting the inpatient data, we have added the data of coronary angiography to this article. In the STEMI group, there were 248 cases of single-vessel disease, 67 cases of two-vessel disease, and 33 cases of three-vessel disease. we found that there was no difference in the number of diseased blood vessels between the two groups, and the number of diseased blood vessels was not related to MACE.

Comment:
Quite unusual… That means that in a period of two years of FUP to have one vessel disease or three vessel disease doesn’t change the prognosis?! In a group of patients, with a mean age of 62y, where 18% of patients has diabetes… How many patient has the criteria for metabolic syndrome? May be a short comment or an explanation has to be take into consideration…

8) The authors choose to assess the consequences of AMI performing an echocardiographic exam in the first 7-10 days after the acute event. The phenomenon of post infarction remodeling is a long one and only a single Echo exam, performed after 7-10 days may not reflects it. So, use of Echo as method the evaluate the extension and the gravity of AMI at 7/10 days is not sufficient… Or is too early… Way we cannot see the echo exam on FUP?
Reply: Thank you very much for the valuable suggestions of the reviewer. We chose to perform an echocardiogram on days 7-10 because most STEMI patients will be discharged after 10 days. Due to patient mobility and cost issues, most patients do not initiate a repeat echocardiographic exam. We will pay attention to echocardiographic data in the subsequent study of AMI. In addition, detailed echocardiographic data of the patients during their hospitalization have been added to the manuscript.

Comment: if we don’t have the possibility to follow the patients in Echo, may be this method should be excluded from the study? HOW can identify the Echo in the first 10 days the patients with high risk, if we performed the PTCA in the first 12 hours?
The big problem arises from the fact that you don’t perform another Echo on FUP…

9) In a group of 348 pts with AMI how many patients have had mitral regurgitation? What about the degree of MR? Does the MR correlates with MACE and FGF21?
Reply: We feel very sorry for the confusion caused by ignoring MR. We have supplemented the data of MR. In the STEMI group, there were 283 patients without MR, 37 patients with mild MR, 12 patients with moderate MR, and no patients with severe MR. In the survival analysis, we did not classify MR, but divided it into presence and absence, and found that MR was related to MACE.

Comment:
Some general considerations and suggestions regarding ischemic MR:
If the AMI is big enough the LV cavity will dilate and a secondary functional mitral regurgitation will develop.
Bigger is the AMI, greater will be the MR.
So, the MR is a very important marker of LV remodeling post AMI.
THIS ASPECT WILL HAVE A NEGATIV IPMACT ON FUP AND MACE.
You’ve mention that the MR was assessed using the Gottdiener & Col. criteria published in J Am Soc Echocardiogr, 17: 1086-119. DOI 10.1016/j.echo.2004.07.013.
I understand / extrapolate that you’ve used the PISA method to quantify the MR.
But I cannot find any criteria in the manuscript to quantify the degree of MR…
A trivial MR has a complete different prognosis compared with a severe MR…
That way on my opinion you have to specify which kind of MR quantification did you used or, at least, WHEN did you take into consideration a significant MR.

10) What about the systolic pulmonary pressure in patients with MACE? Does the pulmonary pressure correlates with MACE and FGF21?
Reply: We are very sorry for not considering the impact of pulmonary pressure on the prognosis. By consulting the inpatient data, we have added the pulmonary artery pressure data to the article. The cox survival analysis found that the pulmonary artery pressure was related to MACE.

Comment: Looking at the echo date (LVEF and pulmonary pressure) seems that all patients are in a good NYHA functional class… Please specify.

11) Why they used only left ventricular end diastolic diameter (LV EDD) and only the ejection fraction to assess the degree of LV remodeling and the level of cardiac reserve after AMI. The LV volumes express much better the phenomenon of post myocardial infarction than LV EDD. Does the LV indexed volumes correlate with MAXCE and FGF21?
Reply: Thank you very much for the valuable suggestions of the reviewer. By consulting the patient’s hospitalized case data, we added more detailed echocardiographic data, such as LAD, IVST, LVPWT, LVESD, LVESV, LVEDV and LV mass. Unfortunately, the STEMI patients were basically lying in bed during their hospitalization, and therefore height and weight data were not available, which prevented the indexing of LVESV, LVEDV and LV mass. However, the cox survival analysis found that the LAD, LVESD, LVESV and LVEF were related to MACE.

Comment: row 90, 91 – “Anthropometric Parameters and Laboratory Examination The baseline characteristics Height / Weight of the participants and history of hypertension…”
If you don’t have the Height and the Weight of the patients, I think that the use Anthropometric Parameters it’s a wrong lexical solution and you have to choose another one… Anyway the Height and the Weight of the patients are very elementary parameters to be missed / excluded from a clinical study…
Is it really impossible for you to have such a basic data?

12) The authors have to clarify how LVEDD diameter was measured: in M mode or in 2DE? And in which view: PS Long Axis? PS Short Axis?
Reply: We are very sorry for not providing detailed echocardiographic measurement methods. We have added it in the text. The LVEDD was evaluated in the parasternal long-axis views in the M mode.

No comment.

13) Consideration: the cardiac reserve is better expressed by the end systolic diameter, and from the study does not appear that they use this parameter (which is by far more characteristic for reduced cardiac reserve). If the authors don’t assess it, they have to specify why. Does the end systolic diameter correlates with MACE and FGF21?
Reply: We are very sorry for not considering the impact of LVESD on the cardiac reserve. By consulting the inpatient data, we have added the LVESD and LVESV data to the article. The cox survival analysis found that the LVESD and LVESV were related to MACE.

No comment.

14) The same consideration for LVEF: end systolic volumes (indexed) correlates better with a poor cardiac reserve, that EF itself. The authors have to publish the indexed volumes of LV cavity and also the methods of calculation of these volumes: may be Simpson?
Reply: We are very sorry for not providing LVESVi and LVEDVi. Unfortunately, the STEMI patients were basically lying in bed during their hospitalization, and therefore height and weight data were not available, which prevented the indexing of LVESV and LVEDV. However, by consulting the inpatient data, we have added the LVESV and LVEDV data to the article.

No comment.

15) Did the authors assess the Left Atrial indexed volumes (in A4C and A2C view)? Does the indexed LA volume correlate with MACE and FGF21?
Reply: I'm sorry that echocardiography in our hospital can't measure left atrial volume yet. However, by consulting the inpatient data, we have added the LAD data to the article. The cox survival analysis found that the LAD was related to MACE.

No comment.

16) If they performed the M mode assessment, what about the calculation LV mass by this modality? There is any correlation with MACE? Usually it is considered that the increased LV mass predicts heart failure. Does, in this group of patients, LV mass correlate with MACE and FGF21?
Reply: Thank you very much for the valuable suggestions of the reviewer. By consulting the patient’s hospitalized case data, we added IVST, LVPWT, LVEDD to the article. LV mass was calculated by the formula 0.8*1.04*[(IVST + LVPWT + LVEDD) 3 - LVEDD 3] + 0.6. Unfortunately, the STEMI patients were basically lying in bed during their hospitalization, and therefore height and weight data were not available, which prevented the indexing of LV mass. The cox survival analysis found that the LV mass was not related to MACE.

No comment.


17) How many patients in 348 AMI patients presented atrial fibrillation, or other arrhythmias at the onset of AMI? What has been the impact of arrhythmias on MACE and FGF21?
Reply: Thank you very much for the valuable suggestions of the reviewer. By consulting the patient’s hospitalized case data, we added the incidence of atrial fibrillation to the article. In the STEMI group, 27 patients presented atrial fibrillation. However, the cox survival analysis found that the incidence of atrial fibrillation was not related to MACE.

No comment.

Reviewer 2 ·

Basic reporting

I consider the manuscript to be written in a clear and unambiguous style, and it respects a relevant structure.

Experimental design

The experimental design respects the ethical standards. The consent in English has been added, and the methods are well described.

Validity of the findings

After revision, more detailed echo parameters were added, and the relationship between FGF21 and MACE is better explained. The fact that the group with SA was not included in the follow-up remains uncorrected, therefore no comparison with this group can me made during this period, but overall the study is well done and the conclusions are well stated and rationale.

---

## Round 0.3 · Minor Revisions

Dear authors, while I acknowledge the detail of your answers in the point-to-point reply, the essence of these answers should be included in the manuscript text. Please be aware that their role is not to convince the reviewers about your point of view, but to integrate some of the criticism.

Therefore:

1. Please add to the manuscript the concern about the limited role of early echocardiography to predict late outcomes (as you responded in the letter). This information should be available for readers.

2. Include in the manuscript (either Methods or Discussion - at the limitations paragraph) the responses to the reviewer regarding the lack of monitoring stable angina patients, constraints in the repetition of troponin I measurements, and other relevant replies that you gave.

---

## Round 0.4 · accepted · Accept

The revised form is now satisfactory. Thank you.